

SciPost Phys. Lect. Notes 32 (2021)

# Local detailed balance

**Christian Maes**⋆

Instituut voor Theoretische Fysica, KU Leuven

⋆ christian.maes@kuleuven.be

## Abstract

We review the physical meaning and mathematical implementation of the condition of local detailed balance for a class of nonequilibrium mesoscopic processes. A central concept is that of fluctuating entropy flux for which the steady average gives the mean entropy production rate. We repeat how local detailed balance is essentially equivalent to the widely discussed fluctuation relations for that entropy flux and hence is at most "only half of the story."

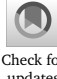

# 1 Introduction

How to construct a statistical mechanical model for nonequilibrium processes that leads naturally to the existence of a stationary state? That question makes the opening line of the paper by Bergmann and Lebowitz (1955) proposing a new modeling framework for the description of irreversible processes [1, 2]. Physically sensible ways for effectively adding the action of reservoirs are indeed central to the study of stationary nonequilibria. We are there at the birth of nonequilibrium statistical mechanics for stationary open systems, driven by the coupling with different spacetime-well-separated equilibrium baths. The resulting stochastic models and identification of currents and entropy flows followed the procedure of what is now called the condition of local detailed balance (LDB).

Making a *viable model* is obviously an art challenging us beyond the mere ensuring to reproduce experimental facts. We want models to be physically motivated, not by their results (alone) but also by their origin and principled derivations. Local detailed balance (the term first appeared in [3,4]) is such a constructive principle. When modeling an open system weakly coupled to equilibrium baths which are well-separated, it makes good sense, and in fact time-symmetry requires, to impose LDB. It has indeed been used as a modeling guide, especially when introducing the problem of current fluctuations; see e.g. Section 2 in [5].

There are naturally two questions about LDB: what it is and when it is. Each of the questions involves a physical context and mathematical framework whereby explanations may vary depending on specific ingredients. Nevertheless there is a general line of answering those questions, which this paper is trying to follow; see also [6–8] for such studies. One should understand that nonequilibrium phenomena are obviously way too diverse to submit to a simple classification or to make an overarching characterization. Processes running under LDB form a much simpler category. That is also why they are at the beginning of studies in nonequilibrium statistical mechanics. Note in that respect that the physical origin of detailed balance is identical to the one for LDB, with detailed balance applying when coupling the system to a single equilibrium bath.
The plan is then as follows. In the next section we give what is meant by local detailed balance, and we repeat what it implies (almost immediately). We start that section with examples to which we return in Section 3. In Section 4 we explain why indeed local detailed balance is a natural condition for a certain class of nonequilibrium processes. We present a derivation and specify the assumptions following the original paper with Karel Netočný, [6]. The main relation and mathematical conclusion is (38), standing for LDB. We end in Section 5 with illustrations of the necessity of the assumptions, bringing us also in the context of multi-channel, active and strongly-correlated processes for counterexamples. While a further reduction (or physical coarse-graining) of a detailed balance process remains detailed balance, that is not true more generally for LDB. At any rate, at best LDB determines only half of the dynamical ensemble for nonequilibria; the rest is also subject to the so called frenetic contribution [9].

# 2 The condition of local detailed balance

We begin with three examples through which appears the nature and importance of LDB. Many similar examples have been discussed at least since [10] in the context of a steady state symmetry for the fluctuating entropy flux in driven systems.

*Example* 2.1 (Generalized Langevin process). A particle of mass one has a one-dimensional position $x_t$ and velocity $v_t$ at time $t$ as it moves through a viscous medium at uniform temper-

ature $T$,

$$\frac{\mathrm{d}x_t}{\mathrm{d}t} = v_t , \tag{1}$$

$$\frac{\mathrm{d}v_t}{\mathrm{d}t} = -\int_{-\infty}^{+\infty} \mathrm{d}s\, \gamma(t-s)v_s + F_t(x_t) + \sqrt{2k_B T}\, \eta_t .$$

The memory kernel $\gamma(t) \geq 0$ for the friction is causal, $\gamma(t) = 0$ for $t < 0$. The forcing $F_t$ is possibly time-dependent or random, and in higher dimensions (or even in one dimension with periodic boundary conditions) it would be easy to make it rotational (non-conservative). The $\eta_t$ is a stationary Gaussian noise-process with zero mean. Such stochastic dynamics (1) is very standard, especially in its Markov approximation. We do not review how it can be obtained from a Hamiltonian or more microscopic beginning. Instead, we use LDB to create physical sense: we wish to specify the time-correlations $\langle \eta_s \eta_t \rangle$ even without recalling the microscopic derivations: the noise correlations are to be determined from LDB.

Noise and friction are caused by the thermal bath which is maintained in equilibrium at temperature $T$, while subject to (supposedly) reversible energy fluxes from the system. From understanding (1) in that way, the entropy flux for a system trajectory $\omega = (x_s, v_s)_s$ equals

$$S(\omega) = -\frac{1}{T} \int \mathrm{d}s\, \dot{v}_s\, v_s + \frac{1}{T} \int \mathrm{d}s\, F_s(x_s)\, v_s . \tag{2}$$

The first term in the right-hand side (a Stratonovich integral) accounts for the kinetic energy difference between the initial and final state of the trajectory, and it represents kinetic energy dissipated in the environment. The second term represents the time-integrated dissipated power (as Joule heat) exerted by the external force. We divide by the bath temperature $T$, to get the change of entropy as dictated by the Clausius relation.

For convenience we deal now with doubly-infinite trajectories $\omega = \big((x_s, v_s), -\infty < s < +\infty\big)$ for the system. They determine the noise-trajectory $\eta_t$ via (1). As the latter is a stationary Gaussian process the path-space probability corresponding to (1) gives weight $\mathrm{P}(\omega)$ to a trajectory $\omega$, proportional to

$$\mathrm{P}(\omega) \propto \exp -\frac{1}{2} \int \mathrm{d}s \int \mathrm{d}u\, \Gamma(u-s)\, \eta_s\, \eta_u , \tag{3}$$

determined by the symmetric kernel $\Gamma(t)$ for which

$$\int \mathrm{d}s\, \Gamma(t-s)\, \langle \eta_s \eta_u \rangle = \delta(t-u) . \tag{4}$$

In the ratio of probabilities (3) we forget possible initial conditions, taking into account the dynamics only.

Now to the crucial condition: LDB amounts to requiring that the log-ratio of the probability of a trajectory to the probability of the time-reversed trajectory equals the entropy flux $S$ from that trajectory:

$$k_B \log \frac{\mathrm{dP}}{\mathrm{d}\tilde{\mathrm{P}}\theta}(\omega) = S(\omega) , \tag{5}$$

where $\theta$ is the time-reversal operator:

$$\theta x_t = x_{-t} , \quad \theta v_t = -v_{-t} , \quad \text{and } \tilde{\mathrm{P}} \text{ refers to using } \tilde{F}_t = F_{-t} . \tag{6}$$

The tilde-process $\tilde{\mathrm{P}}$ uses the time-reversal of the non-magnetic external force. On the other hand, the entropy flux $S$ to be used in (5) is given by (2): we thus demand

$$k_B \log \frac{\mathrm{dP}}{\mathrm{d}\tilde{\mathrm{P}}\theta}(\omega) = -\frac{1}{T} \int \mathrm{d}s\, \dot{v}_s\, v_s + \frac{1}{T} \int \mathrm{d}s\, F_s(x_s)\, v_s . \tag{7}$$

That relation (it is the condition of LDB) is purely dynamical. It is not a definition but a requirement that we impose because microscopic time-reversibility commands it; see Section 4.

To verify (7) one must substitute the $\eta_s$ and $\eta_u$ in (3) from the second line in (1). It is shown in the Appendix of [11] that for the model (1), LDB (7) holds whenever the time-symmetric part of the friction equals the noise-covariance,

$$\langle \eta_s \eta_t \rangle = \frac{1}{2} \left[ \gamma(t-s) + \gamma(s-t) \right] = \frac{1}{2} \gamma(|t-s|). \tag{8}$$

Note that this has become independent of the driving $F_t$, expressing only the thermal equilibrium of the bath: (8) assures (7) for all non-magnetic driving forces $F_t$. The equality (8) is traditionally called the Einstein relation and in the given context is essentially equivalant with LDB. The conclusion is that LDB (= requiring (5)) asks that $\langle \eta_s \eta_t \rangle = \frac{1}{2} \gamma(|t-s|)$ be imposed on the noise-covariance of the dynamics (1).

*Example* 2.2 (Lattice gas). We consider a system of particles in contact with a thermal bath at temperature $T$ and open to particle exchanges at its boundary (only). The isothermal formulation, as in the previous example, is for simplicity. As a model we attempt a Markov jump process where trajectories $\omega$ concern the evolution of occupation variables $x(i)$; see also [12]. The particles are thought to sit on the sites $i$ of the interval $\{1, 2, \ldots, N\}$. There is at most one particle, $x(i) = 0, 1$ (vacant or occupied) per site. The bulk dynamics is governed by an energy function $E(x)$ including interactions and self-potentials. The internal hopping of particles to neighboring empty sites is coded in the transition $x \to y$ and causes the thermal bath to change its energy with $E(x) - E(y)$. That gives the first contribution to the entropy flux. Secondly, particles can enter and leave the system, at both ends $i = 1, N$ from contact with particle baths at chemical potential $\mu_\ell$ and $\mu_r$ respectively, at the same temperature $T$. All that must be summarized properly, i.e., made compatible with the microcopics, in the Markov transition rates $k(x, y)$; that is where LDB will enter.

We take the above very literally: writing $\mathcal{N}_\ell, \mathcal{N}_r$ for the number of particles in the left, respectively right reservoirs, the total change of entropy in the environment is

$$S(\omega) = -\frac{1}{T} \Delta E - \frac{\mu_\ell}{T} \Delta \mathcal{N}_\ell - \frac{\mu_r}{T} \Delta \mathcal{N}_r, \tag{9}$$

with $\Delta$ indicating the total change in energy $E$ and in particle numbers $\mathcal{N}_\ell, \mathcal{N}_r$ as function of the system trajectory $\omega$. In other words, the right-hand side of (9) is the total entropy change in the environment which consists of equilibrium baths; the transitions in the system appear as reversible transformations to which we apply the Clausius definition of entropy. What is assumed in that interpretation and expressed in (9) is that this entropy flux can be read from and is determined by the trajectories of the system. Those trajectories $\omega$ are piecewise constant, with transitions $x \to y$ which each contribute an entropy change (per $k_B$)

$$s(x, y) = \beta(E(x) - E(y)) - \mu_\ell \beta (x(1) - y(1)) - \mu_r \beta (x(N) - y(N)) \tag{10}$$

to $S(\omega)$. LDB here amounts to the requirement that the Markov transition rates satisfy

$$\log \frac{k(x, y)}{k(y, x)} = \text{entropy flux per } k_B \text{ during } x \to y = s(x, y), \tag{11}$$

given from the previous line. That formulation does not require locality of the transition rates, but obviously that may very well be an extra requirement. In fact, LDB assumes that for every transition $x \to y$ only one of the equilibrium reservoirs in the environment gets affected. We conclude from (11) that LDB determines the antisymmetric part in the transition rates.

*Example* 2.3 (Curie-Weiss). The two examples above are common in the sense that the dynamical variables of the system are all equivalent *a priori*. They define so to speak the microscopic level in the description of the system dynamics. We add here a simple example, in fact for a detailed balance process which is a special case of LDB, to illustrate what happens when the system condition is not microscopic in that sense. That happens for example in the case of mean-field analysis, as we now show for a purely dissipative relaxation to equilibrium. It is a version of the open Ehrenfest model which is equivalent to the kinetic Ising model with mean field interaction.

Consider $N$ Ising spins $\sigma_k = \pm 1$, and their magnetization $x(\sigma) = \frac{1}{N} \sum_k \sigma_k$. The spins interact at inverse temperature $\beta$ via a potential $V_N(x(\sigma))$ that depends only on the magnetization. What is a physically motivated way to specify a Markov process on the $x-$variables? We need transition rates $k(x, y)$ for example with $y = x \pm 2/N$ to allow a single spin flip (only). We invoke detailed balance, very similar to (11) except that we should take into account the entropy of the state $x$ itself, and not only the entropy flux: we require

$$\frac{k(x, x + 2/N)}{k(x + 2/N, x)} = \frac{1-x}{1+x+2/N} e^{\beta\left(V_N(x) - V_N(x + 2/N)\right)}. \tag{12}$$

That indeed gives the equilibrium distribution $\nu_N(x) \propto \binom{N}{N\frac{x+1}{2}} e^{-\beta V_N(x)}$ but that is not the main point here. What matters and what is the essence of detailed balance and LDB alike is that (12) expresses

$$k_B \log \frac{k(x, y)}{k(y, x)} = \text{ entropy production in } x \to y,$$

where the entropy production now incorporates the change in internal degeneracy $g_N(y) - g_N(x)$

$$g_N(x) = \log \binom{N}{N\frac{x+1}{2}}$$

as well, in addition to the heat $V_N(x) - V_N(y)$. That obviously plays also in the relaxational behavior. Assuming $V_N(x) = NV(x)$ and taking the limit $N \uparrow \infty$, we get the relaxation

$$\dot{x} = \chi(x) \sinh\left(-\beta w'(x)\right), \tag{13}$$

with $w(x) = V(x) - \beta^{-1} h(x)$ and where $h$ is a mixing entropy (per $k_B$)

$$h(x) = -\left[\frac{1+x}{2} \log\left(\frac{1+x}{2}\right) + \frac{1-x}{2} \log\left(\frac{1-x}{2}\right)\right]. \tag{14}$$

The prefactor $\chi(x) = 2a(x)\sqrt{1-x^2} \geq 0$ is a susceptibility which is also determined by the *symmetric* prefactor $a(x) = \sqrt{k(x, x + 2/N) k(x + 2/N, x)}$ and *not* by detailed balance alone. The free energy $w$ is decreasing in time following

$$\dot{w}(x) = w' \dot{x} = w' \chi(x) \sinh(-\beta w'(x)) \leq 0, \tag{15}$$

making (13) possibly the simplest example of a nonlinear gradient flow; see also [13].

## 2.1 General set up

We proceed with a general description of what is involved in LDB. Before anything else, let us remember that we are up to formulate a requirement for physical modeling of *certain* nonequilibrium systems. Practically speaking, we are given a system driven by contacts with various (ideal) equilibrium baths, and we want to say what model requirement is physically sound. There are then two types of things we need to consider (and combine): (1) the probability of

system-trajectories on some level of reduction, and (2) the possibility of expressing the change of entropy in the environment as a function of system trajectories.

(1) Subsystems, open systems or reduced variables do not necessarily come with an autonomous dynamics. The environment and hidden degrees of freedom are playing their role in the state updating. Depending on the nature of that environment and on the type of reduced description, we end up with a stochastic description and system dynamics from incorporating a fluctuating environment at some point in the past. Whatever is the case, we assume here that we have a dynamical ensemble for the system, meaning a family of possible or allowed trajectories $\omega$ and relative weights or probabilities to express their plausibility. We also assume the presence of a time-reversal operator $\theta$ defined on trajectories, an involution which leaves the family of allowed trajectories invariant. Obviously, what to take for $\theta$ depends on the physical interpretation of the dynamical variables. For odd variables, $\theta$ includes the kinematical time-reversal $\pi$. An example is (6) where $\pi$ flips the sign of the velocities.
Those ingredients allow to define the log-ratio of probabilities of forward to backward trajectories:

$$R(\omega) := \log \frac{P[\omega]}{\tilde{P}[\theta\omega]} . \tag{16}$$

As in (6) the tilde in the denominator for $\tilde{P}$ indicates that we also reverse time-dependent forces or protocols for changing parameters. We admit that the notation (39) is decided by simplicity while the mathematical precision depends on the type of path-space measure. In the context of stochastic processes, the existence and description of $R$ depend on the application of the Radon–Nikodym theorem, also known as Cameron-Martin and Girsanov theorems in general stochastic calculus, [14]. To make sure that (16) only depends on the dynamics (updating rules of the system state or variables), the probabilities are understood as conditional probabilities given the initial state. In still other words, the dynamical ensemble is characterized from an action $\mathcal{A}$, only depending on the dynamics, and $R$ is the dynamical source of time-reversal breaking,

$$R(\omega) = \mathcal{A}(\theta\omega) - \mathcal{A}(\omega), \qquad P[\omega] \propto e^{-\mathcal{A}(\omega)}. \tag{17}$$

(2) We imagine the environment as consisting of various and separate thermodynamic equilibrium reservoirs. In that way (only), it makes sense to speak about the entropy of the environment, and about changes in thermodynamic entropy as the sum of the changes in entropy in each bath. Because of the coupling with the system, that total change of entropy $\Delta S$ over some time-interval clearly depends (also) on the specific trajectory $\omega$ the system has taken during that time. In fact we assume that $\Delta S = \Delta S(\omega)$ is a function of the system trajectory (only).

The condition (or, requirement) of local detailed balance states that

$$\begin{aligned} k_B R(\omega) &= \Delta S(\omega), \qquad \text{i.e.} \\ \Delta S(\omega) &= k_B \log \frac{P[\omega]}{\tilde{P}[\theta\omega]} \end{aligned} \tag{18}$$

for all allowed trajectories $\omega$ over a time-span over which the change of entropy is evaluated. As a reminder, in the case where the system trajectory concerns variables with different degeneracies, as in Example 2.3, we need to include the change of the system entropy. In many cases it can be ignored as the system description naturally involves *a priori* equivalent states.

It must again be emphasized that (18) is *not* a definition but a requirement and property shared by physical models accommodating a nonequilibrium driving induced by couplings with

equilibrium reservoirs. Proving its validity starting from first principles is not straightforward at all and needs further assumptions, which is discussed in Section 4.

## 2.2 Fluctuation relations

If a model satifies LDB in the form (18), some further relations follow immediately, even often useful for understanding the origin of the second law; see e.g. [15]. Indeed, suppose we now do include initial probabilities $\mu$ and $\nu\pi$, indicated as $P_\mu$ and as $\tilde{P}_{\nu\pi}$:

$$\frac{1}{k_B}\Delta S(\omega) + \log\frac{\mu(x_0)}{\nu(x_t)} = \log\frac{P_\mu[\omega]}{\tilde{P}_{\nu\pi}[\theta\omega]}, \tag{19}$$

where we put

$$P_\mu[\omega] = \mu(x_0)P[\omega], \qquad \tilde{P}_\nu[\theta\omega] = \nu(x_t)\tilde{P}[\theta\omega],$$

as we simply think of a time-interval $[0, t]$ for the trajectories $\omega = (x_s, s \in [0, t])$. As shown over the period 1999–2003 in [6, 10, 16, 17] , the LDB identity (19) is essentially the mother of all relations that go under such names as detailed, integrated, local, steady state or transient fluctuation theorems for the entropy flux and that are often associated with the names of Crooks [18], Jarzynski [19] or Gallavotti and Cohen [20]. For the latter we deal with smooth dynamical systems and the Gallavotti-Cohen fluctuation symmetries concern the phase space contraction rate; see [20–23] for references. Obviously, extra assumptions and nontrivial mathematical arguments are needed for the precise mathematical formulation of (19) as a large deviation result, especially when taking the limit $t \uparrow \infty$ and/or when the state space is unbounded. We will not review that here.

## 3 Classified examples

As a matter of further illustration, we start with two classes of Markov processes realizing (18).

## 3.1 Markov jump processes

Denoting states by $x, y, \dots$ the transition rate for a jump $x \to y$ can be parametrized as

$$k(x, y) = a(x, y)\, e^{s(x,y)/2}, \tag{20}$$

with symmetric activity parameters $a(x, y) = a(y, x) = \sqrt{k(x, y)k(y, x)}$ and antisymmetric driving

$$s(x, y) = -s(y, x) = \log\frac{k(x, y)}{k(y, x)}.$$

Under LDB, from (18) and assuming all states are equivalent, the $s(x, y)$ represent (discrete) changes of entropy per $k_B$ in the equilibrium environment. The environment is imagined to consist of spatially well-separated equilibrium baths, each having fast relaxation. Energy, volume or particles are exchanged with one of the various baths during the system transition $x \to y$. Those jumps make the discontinuities in the trajectories which for the rest are constant. The variable entropy flux in the environment is

$$S(\omega) = k_B \sum_\tau s(x_{\tau^-}, x_\tau), \tag{21}$$

where the sum is over the jump times in the (system) trajectory $\omega = (x_\tau, 0 \le \tau \le t)$ and $x_{\tau^-}$ is the state just before the jump to the state $x_\tau$ at time $\tau$. The nonequilibrium process runs as

if locally each transition or each local change in the state (in energy, particle number, volume or momentum) is in contact with one well-defined equilibrium reservoir.

This scenario got realized in the Example 2.2 but the simplest example is of course that of a continuous-time random walk on the one-dimensional lattice. The transition rates to hop to the right, respectively to the left, are $k(x, x+1) = p$ and $k(x, x-1) = q$. In a typical scenario each site $x$ corresponds to a cell of length $L$, repeated periodically and a driving force $F$ works on the walker. When that work gets dissipated instantaneously into a thermal environment (Joule heating) at temperature $T$, the corresponding change in entropy in the bath is $FL/T$. LDB then demands that we put $p/q = \exp[FL/k_B T]$.

## 3.2 Overdamped diffusions

Example 2.1 is an underdamped diffusion. We continue with a calculation for a standard over-damped Markov diffusion. A Brownian particle with position $\vec{r}_t = (r_t(1), r_t(2), r_t(3)) \in \mathbb{R}^3$ moves following

$$\dot{\vec{r}}_s = \chi \vec{F}(\vec{r}_s) + \sqrt{2k_B T \chi} \, \xi_s, \tag{22}$$

with $\vec{\xi}_s$ begin a standard white noise vector, and where the mobility $\chi$ is a constant positive $3 \times 3$−matrix.

To compute the probability of a trajectory we remember that $\vec{\xi}_s, s \in [0, t]$ is (formally) a stationary Gaussian process whose weights are the exponential of minus the quadratic form

$$\frac{1}{2} \vec{\xi}_s \cdot \vec{\xi}_s = [\dot{\vec{r}}_s - \chi \vec{F}(\vec{r}_s)] \cdot \frac{1}{4k_B T \chi} [\dot{\vec{r}}_s - \chi \vec{F}(\vec{r}_s)]. \tag{23}$$

Note however that (23) must be Itô-integrated to obtain the action $\mathcal{A}$ for (17). We can change to Stratonovich-integration, from the identity

$$\int_0^t \vec{G}(\vec{r}_s) \circ d\vec{r}_s = \int_0^t \vec{G}(\vec{r}_s) \, d\vec{r}_s + k_B T \int_0^t (\chi \nabla) \cdot \vec{G}(\vec{r}_s) \, ds, \tag{24}$$

which writes the Stratonovich-integral on the left-hand side in terms of the Itô-integral (first term on the right-hand side). What is important is to remember that the Stratonovich-integral $\int_0^t \vec{f}(\vec{r}_s) \circ d\vec{r}_s$ is anti-symmetric under time-reversal. Therefore, the result for the antisymmetric part (17) in the action is

$$R(\omega) = \beta \int_0^t d\vec{r}_s \circ \vec{F}(\vec{r}_s). \tag{25}$$

It is indeed the Joule-heat divided by $k_B T$. When $\vec{F}(\vec{r}) = -\nabla V$ is conservative, then (25) becomes a time-difference, $R(\omega) = h\beta[V(\vec{r}_t) - V(\vec{r}_0)]$. In any event, we recognize the entropy flux $R = S$ and obviously the Einstein relation in (22) was crucial again. In the opposite direction, we would find that Einstein relation between friction and noise by imposing LDB.

## 3.3 Microcanonical ensemble and detailed balance

The condition of detailed balance ultimately expresses time-reversal invariance of the micro-scopic system (microscopic reversibility) in the microcanonical ensemble. The microcanonical ensemble gives equal probability to all phase-space points on the constant energy-surface. Therefore the Boltzmann entropy itself is giving the weight of a condition:

$$k_B \log \text{Prob}_{mc}[x] = \text{entropy}(x),$$

for probabilities in the microcanonical ensemble. Here $x$ refers to a phase space region, or better we can consider a physically coarse-grained trajectory $\omega$ and then

$$\text{Prob}_{mc}[\omega] = \text{Prob}_{mc}[\theta \omega],$$

by time-reversal invariance. Therefore, the conditional probabilities satisfy

$$\frac{\text{Prob}_{\text{mc}}[\omega \,|\, \omega_0 = x]}{\text{Prob}_{\text{mc}}[\theta \omega \,|\, \omega_t = y]} = \exp \frac{1}{k_B} \{\text{entropy}(y) - \text{entropy}(x)\}. \tag{26}$$

The logarithm of the ratio of transition rates is given by the change of entropy; that is the condition of detailed balance. From here begins the general argument of [6] explained in the next Section that leads to an understanding and derivation of LDB (18), as it is used in nonequilibrium models. The logic and input (in the next section) will be the same as the one leading to (26) but in Section 4.2 the entropy change will refer also to the environment and will depend on the full system trajectory (and not only on its endpoints as it does for detailed balance).

# 4 Derivation of local detailed balance

We follow [6] but in a less detailed way.

## 4.1 Closed systems

On a fundamental level we start from a closed and isolated mechanical systems. Given are $N$ point particles in a volume $V$, in terms of their positions $(q_1, \dots, q_N)$ and momenta $(p_1, \dots, p_N)$ as canonical variables. The phase space point $x = (q_1, \dots, q_N; p_1, \dots, p_N)$ undergoes a Hamiltonian dynamics, $x \to \varphi_t(x)$ in time $t$ for flow $\varphi_t$ on phase space. We assume conservation of energy which means that the motion remains on the energy shell $\Omega_{\mathcal{E}}$ of the initial energy $\mathcal{E}$. Moreover the dynamics is reversible in the sense that there is an involution $\pi$ (kinematical time-reversal, flipping the sign of all momenta) for which

$$\pi \varphi_t \pi = \varphi_t^{-1}. \tag{27}$$

The reversed motion can be obtained by flipping the momenta and applying the original forward dynamics. Finally, the phase volume is preserved, $|\frac{d\varphi_t(x)}{dx}| = 1$, meaning that the Jacobian determinant is one for the change of variables induced by the Hamiltonian flow (Liouville theorem). This is about everything we need for the microscopic dynamics: the conservation of energy, the time-reversibility and the volume-preserving of $\varphi_t$. We write $f := \varphi_\delta$ for the Hamiltonian flow over a fixed, possibly small time $\delta$, and we denote $|M|$ for the phase space volume (Liouville measure) of a region $M \subset \Omega_{\mathcal{E}} : |f(M)| = |M|$ so that $\pi f \pi = f^{-1}$.

Let us move to a description in terms of *reduced* variables. They specify phase space regions $M \subset \Omega_{\mathcal{E}}$. Reduction means that we introduce a level of description where the differences $x \neq y$ in $\Omega_{\mathcal{E}}$ do not count for $x, y \in M$ in the same region. Such a reduced description of the microscopic phase space means to introduce a physically inspired partition of $\Omega_{\mathcal{E}}$. We assume therefore quite generally a map

$$M : \Omega_{\mathcal{E}} \to \mathcal{M} : x \mapsto M(x),$$

where $\mathcal{M}$ is the partition of $\Omega_{\mathcal{E}}$ defining the reduced description. An extra assumption is that when $M(x) = M(y)$ then also $M(\pi x) = M(\pi y)$, which means that the reduced description is compatible with the kinematic time-reversal. We are now ready to look at (reduced or coarse-grained) trajectories $\omega = (M(x), M(fx), \dots, M(f^n x))$ in $\mathcal{M}$, giving the reduced states at times $0, \delta, 2\delta, \dots, n\delta$ as generated by the Hamiltonian flow $f$. They are called the *possible*

trajectories and we only consider those. Note that when $\omega = (M_0, M_1, \ldots, M_n)$ is a possible trajectory in $\mathcal{M}$, then its time-reversal

$$\theta \omega := (\pi M_n, \pi M_{n-1}, \ldots, \pi M_0)$$

is a possible trajectory as well because for all $j = 0, \ldots, n$, if $M_j = M(f^j x)$ for an $x \in \Omega_\mathcal{E}$, then $\pi M_j = M(\pi f^j x) = M(f^{n-j} y)$ for $y = f^{-n} \pi x \in \Omega_\mathcal{E}$. That follows directly from the time-reversibility (27).

Next, we look at the probability of a trajectory $\omega = (M_0, M_1, \ldots, M_n)$. Clearly, $\mathrm{Prob}[\omega]$ needs to measure the probability of drawing an $x \in \Omega_\mathcal{E}$ for which $M(f^j x) = M_j$ at each $j = 0, 1, \ldots, n$. In other words, we look at the region

$$\{x \in \Omega_\mathcal{E} : M(f^j x) = M_j \text{ for all } j = 0, 1, \ldots, n\} = \bigcap_{j=0}^{n} f^{-j} M_j \subset M_0$$

and

$$\mathrm{Prob}[\omega] = \mathrm{Prob}[x \in \bigcap_{j=0}^{n} f^{-j} M_j].$$

We still need an initial condition to define Prob. For that we choose a probability distribution $\mu$ on $\mathcal{M}$, giving weights to the different elements of the partition making the reduced description, and within each $M \in \mathcal{M}$ we use the uniform (Liouville) measure:

$$\mathrm{Prob}_\mu[\omega] = \mathrm{Prob}_\mu[x \in \bigcap_{j=0}^{n} f^{-j} M_j] = \mu(M_0) \frac{|\bigcap_{j=0}^{n} f^{-j} M_j|}{|M_0|}. \tag{28}$$

It means that the probability of a trajectory is first decided by the probability $\mu(M_0)$ of the initial reduced state and then we use the microcanonical ensemble to estimate the fraction of micro-states within $M_0$ that give the required trajectory.

Using (28) we have the following immediate consequence for the log-ratio of probabilities of the forward versus backward trajectory. Suppose $\omega = (M(x), M(f x), \ldots, M(f^n x))$ and $t = n\delta$,

$$\log \frac{\mathrm{Prob}_\mu[\omega]}{\mathrm{Prob}_{\mu\pi}[\theta\omega]} = S(\varphi_t x) - S(x) + \log \frac{\mu(M x)}{\mu(M(\varphi_t x))}, \tag{29}$$

where $S(x) := \log |M(x)|$ is the usual Boltzmann entropy. That can already be compared with (19). In particular, for all times $t$, the conditional probabilities in the microcanonical ensemble of Section 3.3,

$$p_t(M, M') := \mathrm{Prob}_{\mathrm{mc}}[M(\varphi_t x) = M' \,|\, M(x) = M]$$

satisfy

$$e^{s(M)} p_t(M, M') = e^{s(M')} p_t(\pi M', \pi M), \tag{30}$$

where $s(M) = \log |M|$ is still the Boltzmann entropy, defined on reduced variables. The equality (30) is the *condition of detailed balance* of (26) for the reduced level of description. It follows by the time-reversal invariance of the Hamiltonian flow on $\Omega_\mathcal{E}$ and from the stationarity of the microcanonical ensemble.

## 4.2 Open systems

Suppose next that the system is open to an environment. We assume that the previous section appplies to the total of system and environment, but we take special care to make the physical coarse-graining compatible with the system $\otimes$ environment set up. The elements of the partition are now written as a couple $(M, E)$, where $M$ is determined by the state of the system, and $E$ is decided by the state of the environment. To start we can keep in mind that the environment is just a single thermal bath and $E$ stands for its energy[1]. The reduced state $M = M(y)$ depends on the positions and velocities of the particles inside the system.

The region $(M, E) = \{x = (y, z) \in \Omega_{\mathcal{E}} : M(y) = M, E(z) = E\}$ collects the phase space points that are compatible with the given $(M, E)$. Conditional on $(M, E)$, we use the microcanonical ensemble for the total system, which results in giving the weight

$$\nu(y, z) = \frac{\mu(My)\,\mu_{\text{env}}(Ez)}{|My|\,|Ez|} \tag{31}$$

to each phase space point $(y, z)$ of the total system, when $\mu$ is the probability law on the partition of the system and $\mu_{\text{env}}$ is the probability law on the partition of the environment. The choice (31) corresponds to a factorized probability law, for system $\otimes$ environment. We use (31) as the inital probability density for the probability of a trajectory $\omega = (M_0, M_1, \ldots, M_n)$ in the system:

$$\text{Prob}_\mu[M_0, M_1, \ldots, M_n] = \sum_{E_0, E_1, \ldots, E_n} \text{Prob}_{\mu \times \mu_{\text{env}}}[(M_0, E_0), (M_1, E_1), \ldots, (M_n, E_n)],$$

$$\text{Prob}_{\mu \times \mu_{\text{env}}}[(M_0, E_0), (M_1, E_1), \ldots, (M_n, E_n)] = \mu(M_0)\mu_{\text{env}}(E_0) \frac{|\bigcap_{j=0}^n f^{-j}(M_j, E_j)|}{|M_0|\,|E_0|}, \tag{32}$$

and for the reversed trajectory

$$
\begin{aligned}
\text{Prob}_{\mu\pi}[\pi M_n, \pi M_{n-1}, \ldots, \pi M_0] &= \sum_{E_0, E_1, \ldots, E_n} \mu(M_n)\mu_{\text{env}}(E_n) \frac{|\bigcap_{j=0}^n f^{-j}(M_j, E_j)|}{|M_n|\,|E_n|} \\
&= \frac{\mu(M_n)}{\mu(M_0)} \sum_{E_0, E_1, \ldots, E_n} \frac{\mu_{\text{env}}(E_n)}{\mu_{\text{env}}(E_0)} \frac{|M_0|\,|E_0|}{|M_n|\,|E_n|} \\
&\quad \times \text{Prob}_{\mu \times \mu_{\text{env}}}[(M_0, E_0), (M_1, E_1), \ldots, (M_n, E_n)]. \tag{33}
\end{aligned}
$$

That is all exact mathematically. We want to make the ratio between (32) and (33) and see if we get the exponential of the entropy flux, as in (18). We need to add further physical assumptions however. For the reservoir we assume a collection of $m$ equilibrium baths which are each kept at fixed temperature, pressure and chemical potential, and each having an equilibrium entropy $s^{(k)}$ which changes through the evolution $E_0 \to E_n$ from $s_0^{(k)}$ to $s_n^{(k)}$:

$$\log|E_n| - \log|E_0| = \frac{1}{k_B} \sum_{k=1}^m [s_n^{(k)} - s_0^{(k)}], \tag{34}$$

by changes in for example energy, particle number or volume, as in Boltzmann's formula. That obviously requires an idealization and thermodynamic limit where we zoom in on the appropriate scales of time and coupling to allow

$$\text{Assumption 1:} \qquad \frac{\mu_{\text{env}}(E_n)}{\mu_{\text{env}}(E_0)} = 1, \tag{35}$$

---

[1]We would need to specify what we mean exactly with the energy of the environment as there is a coupling (with energy exchange) between system and bath. From here it is natural to assume that the coupling is short-ranged, weak and limited to a spatial boundary of the system. Then, the notion of energy is defined up to boundary corrections. In specific cases, one can of course try to be more precise.

meaning to say that the reduced variables $E$ can change alright from some initial $E_0$ to $E_n$ but that remains unnoticeable for $\mu_{\text{env}}$. For example, if there is an $E$ so that for times $j = 0, \ldots, n$, $(E - E_j)/\sqrt{V} \to 0$ in the volume $V$ of the environment while $\mu_{\text{env}}$ is constant around $E \pm \sqrt{V}$, then (35) holds.

With Assumption 1 and from (34) we have arrived at

$$
\begin{aligned}
\log \frac{\text{Prob}_\mu[M_0, M_1, \ldots, M_n]}{\text{Prob}_{\mu_t \pi}[\pi M_n, \pi M_{n-1}, \ldots, \pi M_0]} \quad = \quad & s(M_n) - s(M_0) + \log \frac{\mu(M_0)}{\mu_t(M_n)} \\
& - \log \left\langle \exp - \sum_{k=1}^{m} [s_n^{(k)} - s_0^{(k)}] \right\rangle,
\end{aligned} \tag{36}
$$

where $s(M) = \log |M|$ is the Boltzmann entropy of the reduced states $M$ in the system.

We are ready for the second and final assumption: the changes in entropy $s_n^{(k)} - s_0^{(k)}$ in each reservoir only depend on the system trajectory $\omega$:

$$
\text{Assumption 2:} \qquad s_n^{(k)} - s_0^{(k)} = J^{(k)}(\omega), \tag{37}
$$

is the time-integrated entropy flux that only depends on the energy, particle number or volume exchanges between system and reservoir $k$. The other reservoirs are spatially separated and their change of entropy is decided by changes in the system. We can make that more explicit: what we need is a rigidity in the separation of equilibrium baths.

Since spatial relations matter we take a snapshot of the system with well-localized places for putting the $N$ particles of the system. How does the energy of a reservoir change? We have a total energy, for phase space points $x$ of the system plus environment,

$$
\mathcal{E} = E_{\text{sys}}(y) + \sum_{k=1}^{m} \left[ U^{(k)}(z_k) + h^k(y, z_k) \right].
$$

Let us see how the energy of the $k$th reservoir $E^{(k)}(y, z_k) = U^{(k)}(z_k) + h^{(k)}(y, z_k)$ changes. if indeed the reservoirs are spatially separated, then the Poisson-bracket between $h^{(k)}$ and $h^{(\ell)}$ vanishes for all $k, \ell = 1, \ldots, m$. Therefore, the energy of the $k$th reservoir is determined by the Poisson-bracket between $h^{(k)}$ and $E_{\text{sys}}$ and between $h^{(k)}$ and $U^{(k)}$. That validates Assumption (37).

Continuing from (36) we have arrived at

$$
\log \frac{\text{Prob}_\mu[M_0, M_1, \ldots, M_n]}{\text{Prob}_{\mu_t \pi}[\pi M_n, \pi M_{n-1}, \ldots, \pi M_0]} = s(M_n) - s(M_0) + \log \frac{\mu(M_0)}{\mu_t(M_n)} + \sum_{k=1}^{m} J^{(k)}(\omega).
$$

We can still rewrite that using conditional probabilities,

$$
\log \frac{\text{Prob}_\mu[M_1, \ldots, M_n \mid M_0]}{\text{Prob}_{\mu_t \pi}[\pi M_n, \pi M_{n-1}, \ldots, \pi M_0 \mid \pi M_n]} = s(M_n) - s(M_0) + \sum_{k=1}^{m} J^{(k)}(\omega). \tag{38}
$$

That is LDB (18) when the change in Boltzmann entropy $s(M_n) - s(M_0)$ for the coarse-grained description of the system is negligible. That is obviously not an assumption but a choice of system-variables, that they all correspond to the same Boltzmann-entropy. It need not hold, exactly as in Example 2.3, in which case we indeed need to add the change of Boltzmann entropy in the system to the total change of entropy appearing in LDB, as we always do e.g. in discussing relaxation of closed and isolated systems.

# 5 Limitations and counterexamples

Local detailed balance fails if the equilibrium reservoirs making the environment are not sufficiently separated, or if the system is directly coupled to a nonequilibrium bath or if the coupling with or between equilibrium reservoirs is too large.

## 5.1 Multiple channels

We may encounter what formally is a reversible process (formally detailed balance) and yet, the log-ratio of the forward to backward rates does not give the (physical) entropy flux. Let us take a specific example. We have a Markov process $(\eta_t, \sigma_t)$ where both $\eta_t, \sigma_t \in \{0, 1\}$, with $\sigma_t$ flipping at a rate $r > 0$ (independent from everything) and $\eta_t$ having transition rates

$$k_{\sigma_t}(0, 1) = \sigma_t \, b + (1 - \sigma_t) a, \qquad k_{\sigma_t}(1, 0) = 1, \tag{39}$$

depending on the state of $\sigma_t$ at time $t$ with parameters $a, b > 0$.

We think of $\eta_t$ as modeling the occupation of a site or *quantum* dot (where there can be at most one particle). Taking $b = e^{\beta \mu_1}, a = e^{\beta \mu_0}$ we have that site in contact with a thermal bath at inverse temperature $\beta$ and with a particle reservoir at chemical potential $\mu_1$ at times when $\sigma_t = 1$ and in contact with a particle reservoir at chemical potential $\mu_0$ at times when $\sigma_t = 0$. That fits perfectly well with LDB for all finite $r$. The entropy flux (per $k_B$) for the $\eta$−transition is simply

$$s_{\sigma_t}(0, 1) = \beta \, \mu_1, \text{ when } \sigma_t = 1 \quad \text{and} \quad s_{\sigma_t}(0, 1) = \beta \, \mu_0, \text{ when } \sigma_t = 0,$$

which is the log-ratio of transition rates (39) indeed.

Let us now look at the limit $r \uparrow \infty$ where the switching of particle reservoirs happens infinitely fast. The $\eta$-process converges to a Markov process with rates

$$k(0, 1) = \frac{1}{2}[a + b], \qquad k(1, 0) = 1. \tag{40}$$

It is a reversible process but there is no LDB because $k(0, 1)/k(1, 0)$ does no longer correspond to an entropy change in the environment. In fact, from a mathematical point of view, insisting on the reversibility, we would say there is no mean entropy production rate in that $r \uparrow \infty$ limiting process, while in fact there is. As a function of the flipping rate $r$, the mean entropy production rate MEP is, for $\beta = 1$,

$$
\begin{aligned}
\text{MEP} &= -\langle [\mu_1 \sigma + (1 - \sigma)\mu_0] J(\sigma, \eta) \rangle \quad \text{for} \tag{41}\\
&\quad J(\sigma, \eta) = k_\sigma(1, 0)\eta - k_\sigma(0, 1)(1 - \eta) \quad \text{so that}\\
\text{MEP} &= -\langle [\mu_1 \sigma + (1 - \sigma)\mu_0][\eta - [\sigma\alpha + (1 - \sigma)\delta](1 - \eta)] \rangle\\
&= -\mu_1(1 + \alpha)\langle \sigma \eta \rangle + \frac{\alpha\mu_1}{2} - \mu_0(1 + \delta)\langle (1 - \sigma)\eta \rangle + \frac{\mu_0 \delta}{2}, \tag{42}
\end{aligned}
$$

where, in terms of the density $\rho := \langle \eta \rangle$, the stationary Master equation gives

$$
\begin{aligned}
\langle \sigma\eta \rangle &= \frac{b + 2r\rho}{4r + 2b + 2}, \quad \langle (1 - \sigma)\eta \rangle = \frac{a + 2r\rho}{4r + 2a + 2},\\
\rho &= \frac{ab + (a + b)(r + \frac{1}{2})}{(a + b)(r + 1) + ab + 2r + 1}, \tag{43}
\end{aligned}
$$

which, when substituted in (42), gives a MEP which is ever increasing in $r$ (unless $\mu_1 = \mu_0$); see Fig. 1.

The $r \uparrow \infty$-process does not satisfy LDB in the given physical context. We cannot exchange the limits $r \uparrow \infty$ and the asymptotic (stationary) time-regime where $t \uparrow \infty$. See [24] for spatially extended examples.

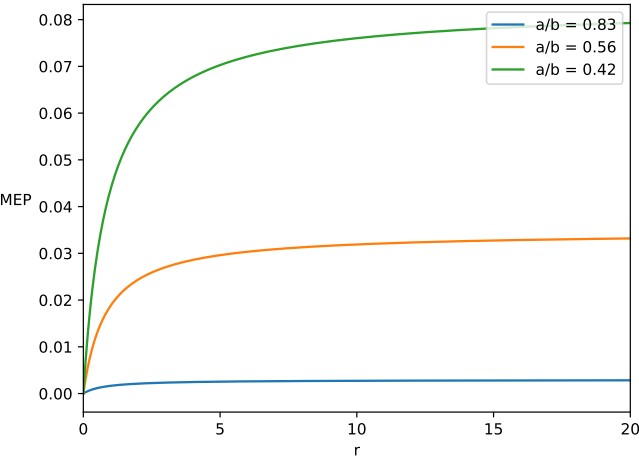

Figure 1: The mean entropy production rate for $a = 0.5$, increasing as function of the rocking rate $r$. The limiting process $r \uparrow \infty$ is a reversible Markov process not satisfying LDB. The value $b$ is decreasing from the top ($b = 1.2$) to the top curve ($b = 0,6$). The MEP vanishes for $a = b = 0.5$. Figure courtesy Simon Krekels.

## 5.2 Strong coupling

If a system is strongly coupled even to one equilibrium reservoir, problems of interpretation appear. Notions of energy transport, heat and dissipation are much harder to define consistently; see e.g. [25]. As we saw in the derivation around Assumption One (35), it is essential to keep the energy fluxes low enough, which is a matter of time-scale and coupling. In fact, already the very notion of energy (of the open system) gets muddy. The so called "Hamiltonian of mean force" does not need to be a true Hamiltonian in the sense of giving rise to well-defined (or, quasi-local) relative energies. Such pathologies are well-documented in the rigorous discussion of statistical mechanics on reduced variables; see e.g. [26].

To understand the dynamical consequence we take a Glauber spinflip dynamics for the low-temperature 2dimensional Ising model at coupling $J > 0$. We have a spinflip rate at site $(i, j)$ given by (for example)

$$c_{ij}(\sigma) = e^{-\beta J \sigma(i,j)[\sigma(i,j+1)+\sigma(i,j-1)+\sigma(i+1,j)+\sigma(i-1,j)]},$$

at which the spin $\sigma(i, j) \to -\sigma(i, j)$. There is detailed balance and the standard nearest-neighbor ferromagnetic Ising model in the plus-phase is an equilibrium distribution. Suppose however we look at the spins $\eta(i), i \in \mathbb{Z}$ on a one-dimensional layer only; e.g. $\eta(i) = \sigma(i, j = 0)$. The process $\eta_t$, as induced from the kinetic Ising model in the plus-phase, is reversible but there is no quasilocal potential to express its detailed balance. In fact, the stationary process for $\eta_t$ is not a Gibbs distribution, [27].

## 5.3 Coupling with nonequilibrium media

If we take a dynamics for an open system that operates under LDB for its interaction with the environment, we can still consider a further reduction or subsystem of it. Suppose indeed that we consider a system with dynamical variables $(x_t, Y_t)$ which jointly undergo a Markov process with LDB. The process $Y_t$ will in general not be Markovian but we can often imagine a further weak coupling for which the $Y_t$ is approximately Markovian. Yet, even so, there is

no reason why the $Y_t$-process will satisfy LDB with respect to is total environment. Counterexamples are easily collected, where a probe interacts with particles under the condition of LDB and its induced fluctuating dynamics does not satisfy the Einstein relation. We refer to [28] for the basic scenario, where the breaking of the Kubo fluctuation–dissipation response is the ultimate origin: even when a process satisfies LDB there is a frenetic contribution to its linear response; see e.g. [29, 30].

The above considerations are applicable to many biological models on mesoscopic scales, including models for active particles. There remains a notion for the distance to equilibrium telling how large are irreversible effects, e.g. using the relative entropy between the forward and the backward evolution probabilities, [16],

$$\mathcal{S}(P|\tilde{P}\theta) = \int \mathcal{D}[\omega] P[\omega] \log \frac{P[\omega]}{\tilde{P}[\theta\omega]}. \tag{44}$$

On mesoscopic scales where the relevant energies are of the order of the thermal energy, theoretical modeling uses stochastic processes that, while case by case possibly very relevant for the discussed biophysics, do however *not always* provide a simple identification of the physical entropy production. In those cases, (44) can still be used as an estimator of the distance to equilibrium but it adds confusion to call it the stationary entropy production (per $k_B$). There is most often not a clear thermodynamic interpretation for the model at hand.

That brings us to the final point, which is the simple message that the symmetric part under time-reversal,

$$D(\omega) = \log P[\omega] \tilde{P}[\theta\omega]$$

(the so called frenesy) is playing a complementary role to (44), crucially important for understanding nonequilibrium physics. When time-reversal symmetry is effectively broken time-symmetric dynamical activity takes a constructive role in selecting the occupation and current statistics alike [9, 31]. That frenetic contribution is kinetic however, and there appears no principle comparable to LDB to specify it.

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
