# Peer review of "Local detailed balance"

_SciPost Physics Lecture Notes, doi:SciPost Phys. Lect. Notes 32 (2021)_

## Round 1 · Referee Report · Anonymous (Referee 1) · 2021-3-11

Strengths

  1. Excellent introduction to the concept of local detailed balance in nonequilibrium statistical physics. The foundations are accompanied by simple examples.

Report

The manuscript by Maes is an excellent introduction to the concept of local detailed balance in nonequilibrium statistical physics. I enjoyed reading the document, and only have a number of minor comments or requests for clarification, for what otherwise is an excellent and clear manuscript. I strongly recommend publication.

Requested changes

  1. p. 2: Wlile -> While

  2. p.3: Non-conservative forces are considered in dimensions D>1. For D=1, can the presence of periodic boundary conditions also facilitate non-conservative forces?

  3. p.5: Equation between (9) and (10) (unfortunately not numbered): aren't there missing contributions for sites 2...N-1?

  4. It might be worthwhile to introduce early on symmetric vs antisymmetric driving and their physical significance and consequences.

  • validity: high
  • significance: high
  • originality: high
  • clarity: high
  • formatting: excellent
  • grammar: perfect

Author:  Christian Maes  on 2021-06-16  [id 1509]

(in reply to Report 1 on 2021-03-11)

Many thanks for the encouraging report. The requested changes are easy to make, except that i am not sure (for 4.) whether i want to bring forward the symmetric vs antisymmetric driving - the paper is about local detailed balance and should not defocus too much.

---

## Editorial Decision

published